# Reliability and Validity of the Emotional Eater Questionnaire in Romanian Adults

**DOI:** 10.3390/nu15010026

**Published:** 2022-12-21

**Authors:** Salomeia Putnoky, Denis Mihai Serban, Ancuta Mioara Banu, Sorin Ursoniu, Costela Lacrimioara Serban

**Affiliations:** 1Department of Microbiology, Discipline of Hygiene, Centre for Studies in Preventive Medicine, “Victor Babes” University of Medicine and Pharmacy, 2 Eftimie Murgu Square, 300041 Timisoara, Romania; 2Department of Obstetrics-Gynaecology, Discipline of Obstetrics-Gynaecology II, “Victor Babes” University of Medicine and Pharmacy, 2 Eftimie Murgu Square, 300041 Timisoara, Romania; 3Department 2, Discipline of Maxillo-Facial Surgery, Faculty of Dental Medicine, “Victor Babes” University of Medicine and Pharmacy, 2 Eftimie Murgu Square, 300041 Timisoara, Romania; 4Department of Functional Sciences, Discipline of Public Health, Center for Translational Research and Systems Medicine, “Victor Babes” University of Medicine and Pharmacy, 2 Eftimie Murgu Square, 300041 Timisoara, Romania

**Keywords:** EEQ, emotional, validation, negative emotions, obesity

## Abstract

Negative emotions and chronic stress trigger abnormal compensatory behaviors known as emotional eating (EE). EE is a well-known mediator for increased body mass index and weight gain. Our aim was to analyze the factor structure and validity and reliability of the Emotional Eater Questionnaire (EEQ) in a sample of 200 Romanian adults with excess weight. Principal component analysis (PCA) was used to assess the construct validity. The mindful eating questionnaire (MEQ) was used to test concurrent validity. Cronbach’s alpha and Spearman correlations were used to analyze internal and external reliability. The socio-demographic characteristics were used as factors for convergent validity. PCA revealed the existence of three major factors, disinhibition, type of food, and guilt, which accounted for 64.9% of the variance. Concurrent validity showed medium to large associations with MEQ (r = 0.650; *p* < 0.001) and a large association with the emotional subscale of MEQ (r = 0.732; *p* < 0.001). Reliability was adequate with Cronbach’s alfa = 0.841 and ICC = 0.775. In a multivariate model, the highest contribution to the EE score was the age (beta = −0.327), followed by feminine gender (beta = 0.321), high levels of perceived stress (beta = 0.215), BMI (beta = 0.184) and lower perceived health status (beta = 0.184). The Romanian version of the EEQ is a reliable and valid tool for measuring emotional eating in adults with excess weight.

## 1. Introduction

For most individuals, acute negative emotions decrease food intake due to increasing associated internal sensations of satiety [1,2]. In some individuals though, negative emotions and chronic stress trigger abnormal compensatory behaviours. Emotional eating, referring to eating in response to negative emotions and stress, is a well-known mediator for increased body mass index and weight gain [3,4]. 

Long-term outcomes of restricted diets for people with excess weight are not satisfactory, with weight regain and lack of beneficial health outcomes mostly being reported [5,6]. Dieters are especially at risk of increasing their food intake in stressful conditions due to disinhibition caused by stress acting on self-control ability [3].

In adults, emotion regulation difficulties are likely to be reported by individuals with excess weight as opposed to individuals with normal weight, along with a lack of planning strategies and emotional awareness, with lower ability to observe, notice, and trust body sensations [7]. Concurrent retrospective and actual high-emotional eater status predicts higher body mass [8]. Difficulty in emotion regulation strategy are reported to affect adherence to treatment in individuals with obesity [9]. Emotional regulation strategies in controlling obesity and food intake are reported as successful in all ages [10,11,12,13], but only current emotional eater status, not former, was associated with weight loss for up to two years [8].

With this in mind, we consider that having a validated tool for the determination of the emotional eater status in individuals with excess weight is a necessity to increase the success of interventions aimed at weight reduction. Our purpose was to determine the validity and reliability of the Emotional Eater Questionnaire (EEQ), developed by Garaulet [14], in a Romanian adult population with excess weight, to be further used in subsequent clinical practice and research. Our hypotheses were that, by using the exploratory factor analysis, a structure similar to the results of Garaulet will emerge and the values of coefficients for internal and external will be greater than the recommended thresholds. 

## 2. Materials and Methods

The Emotional Eater Questionnaire was developed by Garaulet [14] and includes 10 items. It was validated in several other countries [15,16,17,18] and used as an assessment tool in several projects focused on eating disturbances [17,19,20,21,22,23,24]. This questionnaire answered the need for an instrument to be used both in research and clinical practice well, and is compliant, easily understood by patients, and interpreted by investigators using the interpretation scale developed by Garaulet. The dimensions of this instrument (Disinhibition, Type of Food, and Guilt) are straightforward and may encourage early and adequate interventions.

First, two independent persons translated the questionnaire from English to Romanian, and then back to English. By comparing the two English versions, minor corrections could be performed in the Romanian draft version. This version was further pretested by a panel of 12 volunteers. Each question was tested for ease of understanding and the maintenance of a sense similar to that conveyed in English, with minor improvements made to some phrases. The final version of the questionnaire in Romanian (Appendix A: Romanian version of the Emotional Eater Questionnaire) was further used for our purposes. An electronic collection tool was used to collect the answers from the respondents. The questionnaire was promoted on social media and enlarged social circles.

A sample of 495 individuals from the general population was recruited, mostly in the online environment between June–July 2022 for a project with a larger purpose running at the Victor Babes University of Medicine and Pharmacy Timisoara, Romania. Though all types of participants responded to the questionnaire, for the point of this validation, only individuals with BMI ≥ 25 kg/m^2^ (*n* = 200) were included. The height and weight for the calculation of the BMI were self-reported.

Construct validity was tested using the exploratory factor analysis, which is the method of choice to test equivalence, especially when validating a questionnaire in another language [25]. We hypothesized that a structure similar to Garaulet’s would emerge.

The recently validated Romanian Mindful Eating Questionnaire by Serban et al. [26] was used to test its concurrent validity. Correlational analysis was assessed between EEQ score and MEQ and EEQ and MEQ’s five subscales: Awareness, Distraction, Disinhibition, Emotional and External. We hypothesized that the correlation between the questionnaires and relevant subscales will be significant.

Internal reliability was assessed by Cronbach’s alpha statistics. External reliability was performed on a subset of participants (*n* = 25), who responded a second time to the EEQ four weeks after the first evaluation, and was assessed by the intraclass correlation coefficient (ICC). We hypothesized that the coefficients would be acceptable with Cronbach’s alpha > 0.7 [27] and ICC > 0.75 [28]. 

In convergent validity using the socio-demographic factors, the EEQ score was tested as an univariate, and then as a multivariate. We hypothesized that EEQ scores would be influenced by demographic variables, as reported in the literature.

Several other variables, besides EEQ and MEQ-specific questions, were collected and further used as factors: gender, age, self-reported weight and height, self-evaluation of health status (with five possible ordinal categories from excellent to poor), self-evaluation of perceived stress level (ordinal categories from 1 to 10, higher scores associated with higher levels of stress) and marital status (nominal with 6 categories).

### 2.1. Ethics Approval

The ethics approval for this research protocol was obtained from the Scientific Research Vice-rectorate of Victor Babes University of Medicine and Pharmacy Timisoara, Romania with number 12722 on 14 June 2022. All subjects included in the study provided informed consent before participation.

### 2.2. Data Management and Data Analysis

IMB-SPSS version 21 was used to process the data. Using the original instruction for scoring provided by Garaulet [14], each item was scored from 0 = Never; 1 = Sometimes; 2 = Generally; 3 = Always. Higher scores are associated with a higher status of emotional eating. MEQ score and scores of the subscales were obtained as recommended by Framson [29].

Body mass index (BMI) was calculated as the fraction of the weight in kilograms divided by the squared height in meters. Some of the demographic variables were re-coded, such as self-evaluation of health status (excellent and very good versus others). The perceived level of stress was re-coded in a dichotomic manner (up to 6 points-low stress versus 7 and over-high stress) and marital status was recorded as with a partner (married or in a relationship) versus without a partner.

The normality of the distribution was tested with the Kolmogorov–Wilson test. Continuous data assuming parametric distribution are presented as means and standard deviations. Categorical data are presented as percentages. For comparing parametric data on a two-category factor, the t-test was used. For the correlation, the Pearson correlation was used. The size effect was obtained for the Pearson correlation by squaring the Pearson r value that was obtained as part of the test. Cronbach’s alpha was used to assess the internal reliability and the intraclass correlation coefficient was used for the external reliability. PCA was used to verify the factor structure of the questionnaire. A linear regression model was built using the EE score as a dependent variable and the socio-demographic characteristics as independent variables.

## 3. Results

Participants had a mean age of 37.9 years with an SD of 11.7 years, ranging from 18 to 78 years. The BMI ranged from 25 to 58.7 kg/m^2^, with a mean of 30.1 kg/m^2^ and an SD of 4.8 kg/m^2^. The feminine gender represents 72.5% (145) of the group, 24.0% (48) of the sample evaluated their health status as excellent or very good and 51.5% reported high levels of stress. The demographic characteristics of the sample are presented in Table 1.

### 3.1. Construct Validity

The EEQ was analysed by principal component factor analysis (PCA) with varimax rotation and Kaiser normalization. The overall Kaiser–Meyer–Olkin measure of sampling adequacy was 0.85. Bartlett’s test for sphericity produced a significant result (*p* < 0.001), indicating that the correlations between items were sufficiently large for PCA. Hence, our preliminary analyses confirmed the appropriateness of PCA for the data. Three factors had eigenvalues over Kaiser’s criterion of 1 (Figure 1). The three factors explained 64.9% of the overall variance. Using the threshold of 0.4 for factor loadings, the items that were clustered in the same components suggest that Factor 1, which includes questions 4, 5, 6, 8, 9, and 10, represents disinhibition, explaining 28.7% of the variance. Factor 2, which includes questions 2, 3, and 4 represents the type of food, and explains 20.1% of the variance. Factor 3, which includes questions 1 and 7, represents guilt and explains 16.1% of the variance. Factor loading after rotation of each item is shown in Table 2.

### 3.2. Concurrent Validity

The concurrent validity of the EEQ was calculated using Pearson’s product-moment correlations with MEQ and the emotional subscale of MEQ as another relevant measure of acceptance. Significant correlation emerged between scores of EEQ and MEQ (r = 0.650, *p* < 0.001) and between scores of EEQ and the emotional subscale of MEQ (r = 0.732, *p* < 0.001).

### 3.3. Reliability

Cronbach’s alpha for the total EEQ was 0.841. For subscales, it ranged from 0.654 in the Guilt subscale to 0.829 in the Disinhibition subscale. The intraclass correlation coefficient was 0.775 for the whole questionnaire and for subscales it ranged from 0.584 in Guilt to 0.780 in Disinhibition (Table 3). 

### 3.4. Convergent Validity

Univariate analysis showed a higher level of emotional eating in females, individuals reporting higher levels of stress and lower levels of health status evaluation. EE scores correlate significantly with BMI and age (Table 4).

Table 5 contains demographic significant predictors of emotional eater score. Female gender, levels of stress, perceived health status, age and BMI were the independent variables that contributed significantly to the model. All but age were influenced positively the EE scores. The highest contribution to EE score is the age (beta = −0.327), followed by gender (beta = 0.321), high levels of perceived stress (beta = 0.215), BMI (beta = 0.184) and lower perceived health status (beta = 0.184).

## 4. Discussion

This study aimed to analyse the psychometric properties of the EEQ in a sample of Romanian adults with excess weight. This validation is part of a larger interest of our research team on obesity and follows the recent validation of the Framson’s Mindful Eating Questionnaire (MEQ) with 28 items in the general population. The Garaulet’s Emotional Eater Questionnaire was chosen to be also validated in Romanian because of its practical use during medical consultations and research, with only 10 items. 

Construct validity results were consistent with findings reported by Garaulet [14], who found that the 10-item EEQ had a good fit for a three-dimensional factor structure: Disinhibition, Type of food, and Guilt (Table 2). While the items gathered in the factors Disinhibition and Guilt in our work are identical to the report of Garaulet [14], the factor Type of food includes item number four: *Do you have problems controlling the amount of certain types of food you eat*, in addition to the two items contained in the original report. While this question is also part of the Disinhibition factor, where the emphasis is the action on control, it also, at the conceptual level, fits in the Type of food factor that includes information about craving for specific foods and the degree of dependence to sweet foods, especially chocolate. By adding this item to the Type of food factor, we consider that it completes the general picture by adding information about the quantity of food. There is proof that the way different cultures understand and respond to a question depends on the conceptualization and operationalization of the specific construct tested [30].

The correlation of EEQ to Romanian MEQ (28) had a medium to large effect size, but the correlation with the emotional subscale of MEQ indicated a large effect size, similar to what Garaulet [14] has reported. 

The Cronbach’s alpha (Table 3) analysis showed that the Romanian EEQ was a reliable tool to be used in this group. The internal consistency of the EEQ factors ranged from 0.65 to 0.83 for Cronbach’s alpha, indicating moderate to high cohesion between the items of each factor for measuring specific emotional eating behaviour. The Romanian EEQ summary scale was 0.84, which is comparable to that reported by Gonzales (0.84) [15], Nashwan (0.753) [16] and Arslantas (0.84) [18]. The test-retest reliability (Table 3) of the Romanian EEQ with a four-week interval showed an adequate agreement between testing with an ICC of 0.78. Three factors (Disinhibition, Type of food, and Guilt) achieved a medium to good reliability ranging from 0.584 for Guilt to 0.780 for Disinhibition.

For convergent validity, four variables were considered factors: gender, perception of health status, perception stress of stress levels, BMI (kg/m^2^) and age (years). These factors were tested in univariate (Table 4) and multivariate analysis (Table 5).

Lately, emotional eating has been extensively used to investigate feeding behaviour during lockdowns in COVID-19 [17,20,22,23,24,31,32,33,34]. In studies investigating the relationship between short-term modification of body weight and emotional eating, the authors found that individuals gaining weight had higher emotional eating scores than those who maintained weight or decreased weight [17,31,32]. During COVID-19 lockdowns, Navarro-Cruz et al. [20] have reported that, although in the univariate analysis, lifestyle and emotional feeding behaviour were the main factors associated with self-reported weight gain of ≥5%, in a random-effects model, the interaction between lifestyle and emotional eating explained 2% of the variances, while 64% were due to lifestyle deterioration. The same study [20] has found an increase in sweet or stuffed cookies and cakes as the main predictor in the weight gain category. In a study performed with women [33] in which the authors asked themselves what the feeding patterns of individuals with high-emotional eating were, the main predictors were high fat intake, number of meals, sugar consumption, energy intake, and fast-food intake frequency, among other demographic variables. Answering the same question in individuals with abdominal obesity, Betancourt-Nunez [35] reported a negative association between the high-emotional eating category and healthy dietary patterns that included fruits, vegetables, olive oil, oilseeds, legumes, fish, and seafood with an OR of 0.53; 95% CI: 0.33, 0.88 and positively with the snacks and fast-food dietary patterns that included sweet bread, breakfast cereal, corn, potato, desserts, sweets, sugar, fast-food with an OR of 1.88; 95% CI: 1.17, 3.03. The connection between consumption of highly palatable foods and the risk of developing a short-time modification of eating behaviour and weight gain [36,37] and substance abuse [38] has been previously established. In individuals with excess weight, cross-sectional studies have reported a higher prevalence of highly palatable food intake [36,39]. 

In a seven-year longitudinal study, Konttinen et al. [40] found that depression and development of obesity and abdominal obesity are linked through a behaviour mechanism represented by emotional eating. Several results underline that EE scores increase as the nutritional evaluation of individuals shifts from underweight to obese in a stepwise manner [22,24,33]. Our study showed that even among individuals with excess weight, an increase in BMI by one unit increases the EE score by 0.23 units (Table 5). 

In comparison to individuals who perceive their health status as high, our study showed that those with low perceived health status increase their emotional eating score by 2.3 points when all other variables are constant in the model. The literature on the direct effect of perceived health status on emotional eating is scarce. One path to explain this connection is through the presence of chronic disease. In a review by Quick et al. [41], which focuses on chronic disease and emotional eating, the authors have uncovered that, associated with some chronic diseases such as type 1 diabetes, cystic fibrosis, coeliac disease, or inflammatory bowels syndrome, there is an increased prevalence of both eating disorders such as anorexia nervosa or bulimia nervosa and disordered eating such as emotional eating, disinhibited eating, restraint eating, night eating and binge eating. Another path to explain the connection between perceived health status and emotional eating is through the positive effect of chronic diseases on stress levels. In a large study that included information on 229,293 adults [42], the authors reported that standardized coefficients of linear regression for stress have increased from beta = 5.58 for one chronic disease to beta = 20.17 for four or more chronic diseases, compared with individuals with no chronic conditions. 

Higher levels of stress are associated with emotional eating, both in our report and shown in several others [33,43,44]. These reports have completed the picture by adding sleep disorders in the connection between stress and emotional eating. In a longitudinal study [40], shorter sleep duration exacerbated the causality pattern between emotional eating and accumulation of excess weight. Nurses assigned to COVID-19 units compared with nurses from non-COVID-19 units were 2.62 times more likely to pertain to higher EEQ categories, but age, nursing experience, and quality of life were not statistically different between the groups [16]. In a randomized controlled trial [45], exposure to similar levels of stress in obese and lean individuals resulted in higher levels of cortisol in obese compared to normal weight individuals, without differences in the food intake in a subsequent meal served under laboratory conditions, but with higher levels of restrained eating measured in the group with obesity. 

In our sample, each one-year increase in age was associated in the multivariate model with a decrease by 0.16 units of the EE score (Table 5). The relationship between age and the degree of emotional eating has been previously reported [46,47,48,49]. Samuel et al. [48] found that expressive suppression, a main emotion regulation technique, had a stronger relationship with emotional eating in younger adults, decreased in middle-aged adults, and was insignificant in older adults. Reappraisal is used by individuals as an early method of the down-regulation of emotions and behavioural expression without impacting the memory. Suppression, by inhibiting the display of inner feelings, decreases only behavioural expression and impairs memory. The extensive use of expressive suppression was found to be an obstacle in emotional expression [50]. 

Our finding is consistent with previous studies [16,23,24,44,49] that document a higher prevalence of emotional eating in women compared to men. In our study, with the female gender, the emotional eating score increases by 4.5 points when controlling for the effects of other variables in the model. These differences between genders can be partly explained by physiological differences [51,52,53] due to the interaction of oestrogen and progesterone across the menstrual cycle in females. 

Since participants volunteered to participate, the bias of selection might be of concern. Our validation was performed on adults with excess weight, similar to the original validation method [14]. Others that used the same questionnaire successfully validated the instrument using individuals with normal range BMI [15,16,20,32] or adolescents [17].

## 5. Conclusions

The Romanian version of the EEQ is a reliable and valid tool for measuring emotional eating in adults with excess weight and could be a relevant tool in our war against obesity. Construct, concurrent and convergent validity, and internal and external reliability were adequate and support the further use of the Romanian version of EEQ to preselect patients with overweight and obesity who are also emotional eaters to participate in specific behavioural therapies that could consolidate the weight loss for a longer time. 

## Figures and Tables

**Figure 1 nutrients-15-00026-f001:**
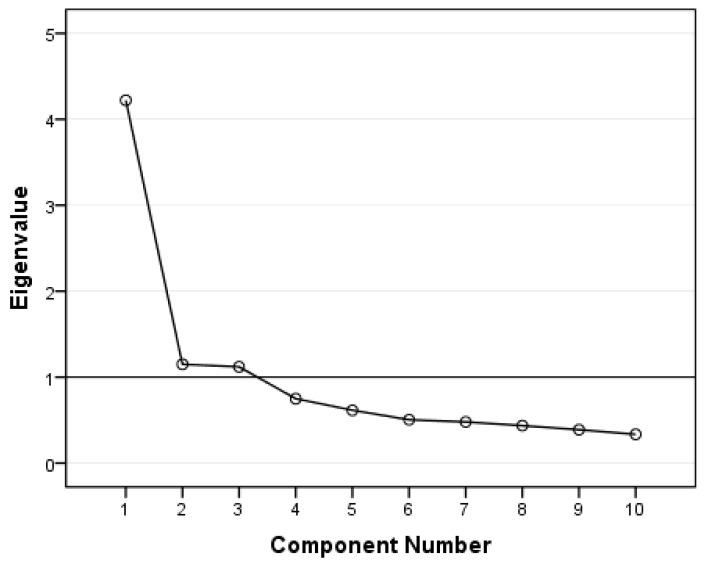
Scree plot of eigenvalues after PCA.

**Table 1 nutrients-15-00026-t001:** Demographic characteristics of the sample (*n* = 200).

Variables		Percentage (Count)
Gender	Males	27.5% (55)
	Females	72.5% (145)
Excellent or very good perceived health status	Yes	24.0% (48)
	No	76.0% (152)
High level of perceived stress	Yes	51.5% (103)
	No	48.5% (97)
Relationship	In a relationship	78.9% (146)
	Single	21.1% (39)
Age (years)		37.9 (11.7) *
BMI (kg/m^2^)		30.1 (4.8) *

* Mean (SD).

**Table 2 nutrients-15-00026-t002:** Factor loadings After Varimax Rotation with Kaiser Normalization.

EEQ	Structure		
	Disinhibition	Type of Food	Guilt
6. Do you eat more of your favourite food and with less control when you are alone?	**0.789**	0.109	0.103
10. How often do you feel that food controls you, rather than you controlling food?	**0.708**	0.314	0.203
5. Do you eat when you are stressed, angry or bored?	**0.703**	0.379	−0.133
9. When you overeat while on a diet, do you give up and start eating without control, particularly food that you think is fattening?	**0.667**	0.207	0.217
8. Do you feel less control over your diet when you are tired after work at night?	**0.648**	−0.020	0.397
2. Do you crave specific foods?	0.107	**0.774**	0.170
3. Is it difficult for you to stop eating sweet things, especially chocolate?	0.162	**0.757**	0.173
4. Do you have problems controlling the amount of certain types of food you eat?	**0.467**	**0.670**	0.040
1. Do the weight scales have a great power over you? Can they change your mood?	0.012	0.299	**0.815**
7. Do you feel guilty when eat “forbidden” foods, like sweets or snacks?	0.361	0.069	**0.783**

* numbers in bold represent factor loadings of items aggregating into each factor.

**Table 3 nutrients-15-00026-t003:** Internal and external reliability.

	Cronbach’s Alpha (*n* = 200)	Intraclass Correlation Coefficient (*n* = 25)
EEQ	0.841	0.775
Disinhibition	0.829	0.780
Type of food	0.709	0.673
Guilt	0.654	0.584

**Table 4 nutrients-15-00026-t004:** Univariate analysis between EE score and selected demographic variables.

Variables		Measures	EE Score	*p*-Value
Gender	Male	Mean (SD)	10.7 (5.5)	<0.001 *
	Female		14.8 (5.7)	
Excellent or very good perceived health status	Yes	Mean (SD)	11.9 (6.0)	0.017 *
	No		14.3 (5.8)	
High level of perceived stress	Yes	Mean (SD)	11.8 (5.5)	<0.001 *
	No		15.4 (5.8)	
BMI (kg/m^2^)		r	0.202	0.004 **
Age (years)		r	−0.190	0.007 **

* *t*-test; ** Pearson correlation.

**Table 5 nutrients-15-00026-t005:** Linear regression for predicting EE score (*n* = 200).

	Unstandardised Coefficients		Standardized Coefficient	95% CI for B	
	B	Standard Error	Beta		
Constant	2.148	2.685		−3.147	7.444
Feminine Gender	4.247	0.825	0.321	2.619	5.875
Lower perceived health status	2.291	0.920	0.166	0.477	4.106
High level of perceived stress	2.543	0.728	0.215	1.108	3.978
BMI (kg/m^2^)	0.228	0.078	0.184	0.075	0.380
Age (years)	−0.165	0.034	−0.327	−0.232	−0.099

Dependent variable: emotional eater score; Independent variables: Gender, perceived stress levels, perceived health status, BMI, age. F(5199) = 16.1, *p* < 0.001, adjusted R square = 0.274.

## Data Availability

The datasets used and/or analysed during the current study are available from the corresponding author on reasonable request.

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
