# Peer review of "Reliability and Validity of the Emotional Eater Questionnaire in Romanian Adults"

_nutrients, 2022, doi:10.3390/nu15010026_

Round 1

Reviewer 1 Report

This paper reports on a straightforward translation (into Romanian) and psychometric check of a questionnaire assessing emotional eating. It follows a very similar procedure to the Garaulet paper that first described this questionnaire 10 years ago. In the main, the paper is well presented and well-written.  A few issues for the authors to consider:

1.    Abstract and later. It’s not clear to me why the authors have chosen to label factor 2 differently to the original: food rather than ‘type of food.’

2.    Introduction. There’s a literature background reviewed that while acceptable, doesn’t seem central to the purpose of the study. The paragraph on childhood and adolescence, for example, has no relevance to the present psychometric determination. Much more relevant is that the questionnaire has been similarly translated and psychometrically assessed in Turkish, Polish and Arabic (noted in line 70).

3.    Lines 64-67. The study purpose is clear. The authors might like to include some of the hypotheses referred to in the methods here e.g. those relating to factor structure (lines 87-88) and reliability (lines 96-98).

4.    Line 85. Was participant weight actually measured or simply self-reported?

5.    Line 188. Substitute interest for preoccupation.

6.    Line 231. With women rather than on women.

7.    Lines 243-249 are speculation. No biological data were reported in this study.

8.    The paragraph on stress (lines 271-289) is similarly unrelated to the data reported in this study. I suggest this is removed.

9.    Line 300. Higher rather than high.

Reviewer 2 Report

Dear Authors,

I identified minor concerns related to your manuscript. For these reasons, I would recommend the paper for publication with minor revisions. 

If you elect to revise and resubmit the paper, please include a letter which details your changes to the paper or your rationale for not making a suggested change. 

-       The manuscript needs English revision by a native English speaker.

Introduction:

-       Lines 48-54: these sentences are not relevant to this paper; you may delete this part.

-       Lines 55-61: please not that difficulty in emotion regulation strategy was reported to affect adherence to treatment in individuals with obesity (DOI: 10.1007/s11695-021-05485-9)

Discussion:

-       Lines 265-266: “Another path 265 to explain the connection between perceived health status and emotional eating is through the positive effect of chronic diseases on stress levels”. It is not clear, it should be a negative effect. 

-       Lines 267-270: this sentence should be elucidated 

-       Lines 303-305: women with obesity reported higher prevalence of depression compared to men (DOI: 10.1007/s00737-021-01123-6)
